# Comparison of development of step-kinematics of assisted 60 m sprints with different pulling forces between experienced male and female sprinters

Roland van den Tillaar ⬤ *

Department of Sport Sciences and Physical Education, Nord University, Levanger, Norway

* roland.v.tillaar@nord.no

## Abstract

The purpose of this study was to compare step-by-step kinematics of normal and assisted 60 m sprints with different loads in experienced sprinters. Step-by-step kinematics were measured using inertial measuring units (IMU) integrated with a 3-axis gyroscope and a laser gun in 24 national level male and female sprinters during a normal 60 m sprint and sprints with a 3, 4, and 5 kg pulling force. The main findings were that using increasing assisted loads resulted in faster 60 m times, as a result of higher step velocity mainly caused by longer step lengths in both genders and by shorter contact times in women. Men had longer step lengths, longer contact times, and shorter flight times than women. However, the assisted loads had a greater effect on women than on men, as shown by their larger decrease in sprint times. These time differences in gender were the result of more and longer duration increases in maximal step velocity with increasing assisted loads for women (70–80% of distance) than men (65–70% of distance). This was mainly caused by shorter contact times, and by more increased step lengths in women compared to men. In terms of practical application, it is notable that employing this approach, when using assisted loads can help athletes to reach higher step velocities and hold this for longer, which may be a training impulse to move the speed barrier upwards.

## Introduction

In many sports like track and field, rugby, American football and soccer, sprint ability is highly important to outrun their opponents to win, get the ball or to score points [1–3]. There are different views on the type of training needed to enhance sprint performance. These views are derived from the force-velocity curve of movement, which in turn is based on the force-velocity curve of muscle contraction described by Hill [4]. Firstly, there is training based upon the principle of an overload of force, e.g. general resistance training and resisted sprints [5–8]. Secondly, there is training based upon the principle of an overload of velocity, e.g. assisted sprints [9–11]. Different ways are used to create this assisted condition like running behind a car, downhill running [12], or by using a pulley system [13, 14]. Some pulley systems, such as

**Funding:** The authors received no specific funding for this work.

**Competing interests:** The authors have declared that no competing interests exist.

elastic bands or being pulled by a partner [9] cannot easy control the amount of assistance exactly over the whole sprint distance. Thereby, these systems are not well suited for research purposes in which assisted load must be controlled to know what exact loads are necessary and for the possibility of replicating the study.

Only a few studies have investigated the kinematics during assisted sprints [9, 10, 13, 15–17]. In assisted runs higher step velocities were found, which were caused by a longer step length and shorter contact times [10, 15–17]. Furthermore, a higher stride rate was found in national level sprinters [9, 17] when performing assisted sprints compared with traditional sprints. However, van den Tillaar and colleagues [10, 15] have only investigated the acute effect of assisted sprints upon kinematics for the first 20–30 m, while Mero and Komi [9] investigated kinematics over a 10 m distance (between 35 and 45 m) and Clark et al. [16] investigated only one stride at 35 m. None of these studies investigated the kinematics for the entire sprint from start until reaching maximal velocity and the acute effect of assisted runs with different loads. Only Clark et al. [16] compared different assisted sprint loads with the normal sprint. However, due to the use of elastic towing it was difficult to control the exact pulling force.

In addition, studies of van den Tillaar and colleagues [10, 15] only investigated running kinematics of female soccer and female team handball players, while Mero and Komi [9], Mann and Murphy [1], and Slawinski et al. [18] only investigated sprint kinematics at different parts of the sprint in experienced male and female sprinters. Slawinski et al. [18] and Brechue [19] found that women achieve peak velocity during a sprint earlier than men and that the peak velocity was lower than in men. This earlier occurrence and lower peak velocity was mainly due to their shorter step length [1, 9], reduced limb/joint stiffness, and their horizontal velocity and step length plateauing sooner than men [19]. When comparing the effect of assisted sprints upon step kinematics between men and women only Mero and Komi [9] investigated this, when using a tightened rubber rope with a runner pulling on average 30–45 N. The use of another partner and a rubber rope causes inaccuracies in pulling force as shown by the variation in pulling force of 30–45 N [16]. Due to this variability in pulling force it is difficult to replicate this accurately in training and for research purposes.

Nowadays, there are also pulley systems, such as the 1080 Sprint™ and DynaSpeed™, which can give a constant active resistance during the whole sprint by using a motor to employ a constant pulling force [20, 21]. These pulley systems can help perform controlled assisted sprints with different loads. As not much is known about the acute effect of assisted sprints with different loads on step kinematics for men and women, more detailed information is needed. The acute effect of several different loads upon sprinting step kinematics for men and women could help in prescribing training programs involving assisted sprints that are specific for gender.

Therefore, the aim of the study was to investigate the effect of different assisted sprint loads on step-by-step kinematics during a 60 m sprint in experienced male and female sprinters. It was hypothesized that step velocity would increase with increased assisted sprint loads and that this increased step velocity would cause a higher maximal velocity. Since a higher maximal velocity is based upon changes in step kinematics it was hypothesised that the assisted sprint loads resulted in longer step length, shorter contact times, and higher step rate as found in earlier studies [9, 10, 16, 17]. In addition, it was expected that 60 m times in women would decrease relatively more than in men, with equal pulling force as women experience more pulling force relative to their body mass.

## Methods

To investigate the effect of different assisted sprint loads in men and women upon step-by-step kinematics during a 60 m sprint, a cross-sectional study design was used in which 24

experienced male and female sprinters performed a normal 60 m and assisted 60 m with 3, 4, or 5 kg of pulling force provided by a motorised pulley system. Kinematic-dependent variables were step velocity, contact and flight time, step length and frequency over the whole 60 m for both genders and all conditions.

## Subjects

Twelve experienced male sprinters (age: 22 ± 6 years, body mass: 74.5 ± 6.8 kg, body height 1.82 ± 0.07 m, personal best on 100 m times of 11.16 (10.27–11.97) s), and twelve experienced female sprinters (age: 22 ± 3 years, body mass: 60.7 ± 5.1 kg, body height: 1.68 ± 0.06 m, with personal 100 m best times of 12.58 (11.74–13.07) s, participated in the present study. They were instructed to avoid any maximal sprint training for at least 48 hours prior to testing. Each subject was informed of the testing procedures and possible risks, and written consent was obtained prior to the study from them and parents or legal guardians when they were under 18 years old. The study complied with current ethical regulations for research, was approved by the Norwegian Center for Research Data and performed according to the Declaration of Helsinki.

## Procedures

Testing was conducted during the indoor season in preparation for the national championships. The subjects were instructed to avoid undertaking any maximal sprint training in the 48 hours prior to testing. After an individualized warm-up of around a 30–45 minutes, each subject performed one 60 m sprint, followed by a 3, 4, and 5 kg assisted 60 m sprints. The warmup was individualized, since the subjects were experienced sprinters who knew best what type of warm-up works for them to attain their greatest sprinting performance and to avoid possible injuries. The assistance of 3, 4, and 5 kg during the 60 m assisted sprints was provided by DynaSpeed (Ergotest Technology AS, Langesund, Norway), which is a portable motorised pulling system to provide assistance/resistance while sprinting. The Dynaspeed was placed at 85m from the subject with the cable attached to the subject on a belt firmly tightened around the waist (Fig 1). The assisted load (3, 4 or 5 kg) was determined and controlled by MuscleLab v10.202.93.5131 software (Ergotest Technology AS, Langesund, Norway). Rest between each trial was 10–12 minutes to avoid fatigue. All subjects had some experience with assisted sprints, but not with this device.

Sprint times were measured with two pairs of wireless photocells placed at height of 1 m (Brower Timing Systems, Draper, UT, USA). Subjects initiated each sprint from a standing start in a split stance, with the lead foot behind a line taped on the floor 0.3 m from the first set

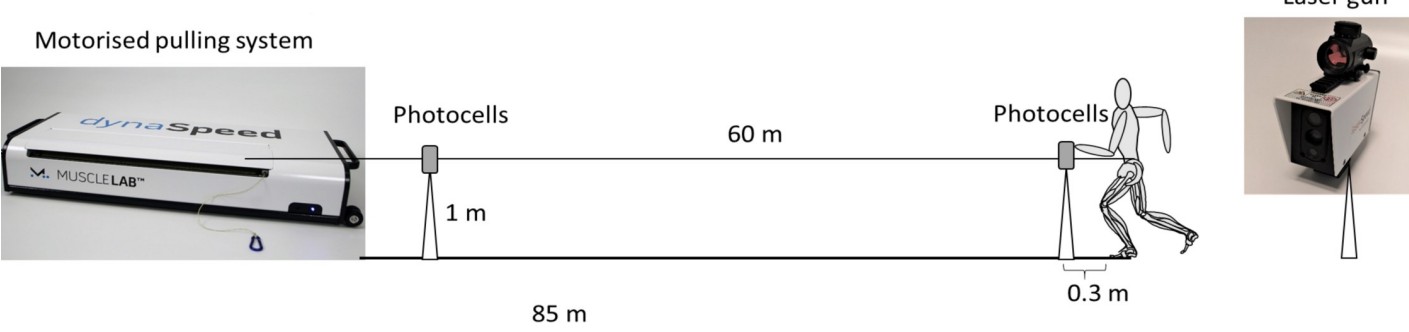

**Fig 1. Testing set up with used equipment.**

of photocells. Distance measurements were recorded continuously during each attempt using a CMP3 Distance Sensor laser gun (Noptel Oy, Oulu, Finland), sampling at 2.56 KHz. Step velocity was automatically calculated by MuscleLab v10.57 (Ergotest Technology AS, Langesund, Norway). Contact and flight times throughout the run were derived from an identified running pattern (Fig 2) using wireless 9 degrees of freedom inertial measurement units (IMU) integrated with a 3-axis gyroscope attached on top of the shoelaces of the spikes of each foot. Sampling rate of the gyroscope was 500 Hz with maximal measuring range of $2000° \cdot s^{-1} \pm 3\%$ (Ergotest Technology AS, Langesund, Norway). This running pattern was determined in an unpublished pilot study that compared contact and flight time data measured with an infrared contact mat over 30 m (Ergotest Technology AS, Langesund, Norway) using the patterns of the angular velocity of plantar flexion/extension (ICC = 0.94). This made it possible to measure contact and flight time directly with the IMU per step and to determine each step and count the number of steps over the whole 60 m. Step frequency and step length were calculated for each step by the formulas: Step frequency = 1 / (contact time + flight time), Step length = velocity $^*$ (contact time + flight time). The velocity for each step was derived from the laser gun, which was the average velocity during one flight and contact time (step). All recordings were synchronized with the MuscleLab v10.57 (Ergotest Technology AS, Langesund, Norway). To account for the difference in number of steps between the conditions and between genders, kinematic data was averaged for each 5% of the total distance. Thus, the average kinematics were calculated in 20 points (every 3 m) for the whole 60 m for each condition.

## Statistical analyses

Normality was tested using the Shapiro-Wilks test of normality. To compare the sprint times between genders and for different assisted sprints, a 2 (gender) x 4 (normal, 3, 4, and 5 kg

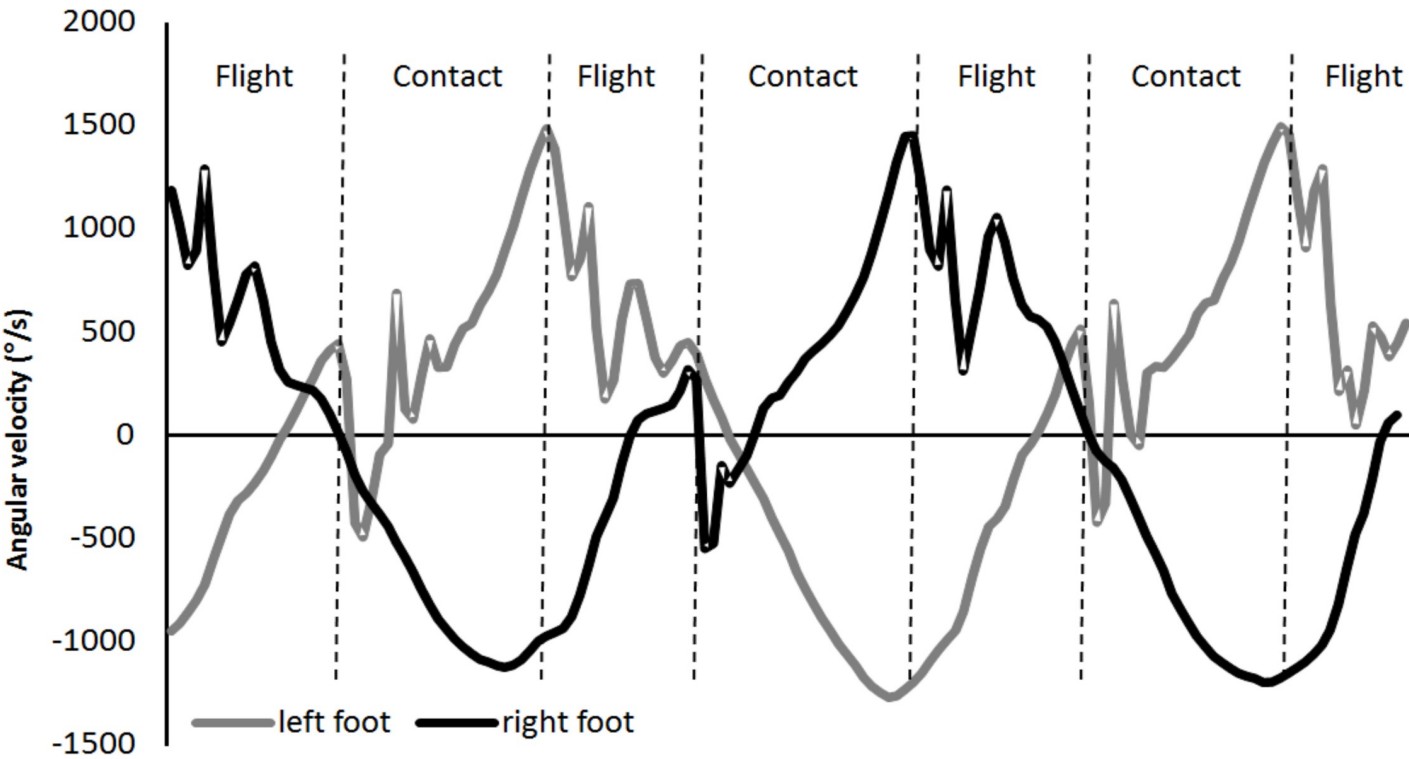

**Fig 2. A typical example of three steps of the raw plantar flexion pattern measured with an IMU attached on the left and right foot with corresponding identified contact and flight time phases.**

assisted sprint) analysis of variance (ANOVA) with repeated measures was used. To assess the effect of different assisted sprints and gender on kinematics, a mixed model 2 (gender) x 4 (normal, 3, 4, and 5 kg assisted sprint) x 20 (percentage of total 60 m sprint distance) ANOVA for each kinematic variable was used. When significant differences were found a two-way ANOVA (Gender and percentage of 60 m sprint) per assisted sprint load was also performed. Post hoc comparisons with least mean differences were performed for pairwise comparisons between each following percentage (5%) of the 60 m sprints. The level of significance was set at $p < 0.05$, and all data are expressed as mean ± standard deviation (SD). Analysis was performed with SPSS Statistics for Windows, version 25.0 (IBM Corp., Armonk, NY, USA). Effect size was evaluated with partial eta squared ($\eta^2$) where $0.01 < \eta^2 < 0.06$ constituted a small effect, $0.06 < \eta^2 < 0.14$ a medium effect, and $\eta^2 \geq 0.14$ a large effect [22].

## Results

The 60 m times decreased from the normal sprint condition significantly with each assisted pulling load (F = 108, $p < 0.001$, $\eta^2 = 0.83$,) for both genders. Relative and absolute decrease in sprint times (Table 1) was more in women compared with men (F = 19.2, $p < 0.001$, $\eta^2 = 0.30$). The number of steps also decreased significantly with increasing assisted loads for both women and men (F = 31.1, $p < 0.001$, $\eta^2 = 0.62$) with no significant difference in decrease between men and women (F = 1.1, $p = 0.34$, $\eta^2 = 0.06$). The men had significantly fewer steps than the women in all conditions (Table 2). Post hoc comparison revealed that the number of steps did decrease significantly from normal to 3 kg assisted sprint and again from 4 kg assisted to 5 kg assisted sprints (Table 2).

### Kinematics

All kinematics were affected by running distance in both men and women (F $\geq$ 21, $p < 0.001$, $\eta^2 \geq 0.64$), while the assisted conditions influenced velocity and step length (F $\geq$ 9.8, $p < 0.001$, $\eta^2 \geq 0.39$) in both genders. On step frequency and contact times only a significant effect in women (F $\geq$ 3.5, $p \leq 0.031$, $\eta^2 \geq 0.34$) was found, but no significant effect on flight time (F = 0.45, $p = 0.72$, $\eta^2 = 0.04$, Table 3). In addition, a gender effect was found for all kinematics in all conditions (F $\geq$ 5.9, $p \leq 0.026$, $\eta^2 \geq 0.25$), except for step frequency (F = 0.3, $p = 0.57$, $\eta^2 = 0.03$). Furthermore, a significant interaction effect was found for running distance and gender for step velocity and flight time for all running conditions (F $\geq$ 5.1, $p \leq 0.001$, $\eta^2 \geq 0.30$), and for the effect of condition*running distance on step velocity and

**Table 1. Mean (±SD) sprint times and absolute and relative change in sprint times per gender.**

| Gender | Normal | 3 kg assisted | 4 kg assisted | 5 kg assisted |
|---|---|---|---|---|
| *Total sprint times 60 m* | | | | |
| Women (s) | 8.17±0.30 | 7.89±0.27 | 7.76±0.28 | 7.62±0.30 |
| Men (s) | 7.23±0.30 | 7.12±0.34 | 7.03±0.34 | 6.93±0.34 |
| *Absolute change with normal 60 m* | | | | |
| Women (s) | | 0.28±0.16 | 0.41±0.17 | 0.55±0.18 |
| Men (s) | | 0.12±0.10 | 0.20±0.07 | 0.30±0.10 |
| *Relative change with normal 60 m* | | | | |
| Women (%) | | 3.39±1.88 | 5.02±1.98 | 6.72±2.15 |
| Men (%) | | 1.62±1.42 | 2.80±1.06 | 4.21±1.36 |

Significant different between men and women in every condition and between conditions on a p<0.05 level.

**Table 2. Mean (±SD) number of steps during the 60 m per gender under each condition.**

| Gender | Normal | 3 kg assisted | 4 kg assisted | 5 kg assisted |
|---|---|---|---|---|
| Women (n)† | 31.6±1.1* | 30.7±0.8 | 30.5±0.9 | 30.2±0.9* |
| Men (n) | 36.1±1.5* | 34.9±1.6 | 34.7±1.7 | 34.0±1.7* |

* indicates a significant difference with all other conditions on a p<0.05 level.

† indicates a significant difference in number of steps with men in each condition on a p<0.01 level.

step length (F ≥ 1.6, p ≤ 0.004, $\eta^2$ ≥ 0.13), and there was also a significant interaction effect of condition*distance*gender on step velocity (F = 1.5, p = 0.015, $\eta^2$ = 0.10).

Post hoc comparisons revealed that men reached a higher step velocity and that they obtained this earlier (at 65–70% of the 60m) than women (75–80% of the 60m) for the different assisted conditions (Fig 3, Table 4). Men had longer step lengths, longer contact times, and shorter flight times than women (Figs 3 and 4). The assisted loads resulted in a higher step velocity with each increased assisted load in both men (+2.1–5.4%) and women (+3.9–8.8%) Fig 3). Step length increased with every assisted load for women (+3–7.9%) and for men between normal and 3 kg (+2.1%) and between 3 and 5 kg assisted loads (+4.8%, Fig 4). Contact times decreased significantly between 3 and 5 kg assisted sprints in women (-3.9%) (Fig 5), while post hoc comparison did not find a significant step frequency difference between conditions in women (Table 3). Step velocity was significantly different between conditions after 15% of the distance for both, while it increased further with larger assisted loads and longer distances for women (70–80% of distance) than men (65–70% of distance). After reaching the maximal step velocity, maximal step velocity decreased again at 75% of the distance (45m) in a normal 60 m sprint in men to no reduction in step velocity with 5 kg assisted loads (Table 4 and Fig 3). In women, step velocity did not significantly decrease after reaching maximal step velocity in the normal and 4 kg assisted sprints, while it deceased at 85 and 90% with 3 and 5 kg assisted loads (Fig 3, Table 4).

**Table 3. Average (±SD) step velocity, step length and frequency and contact and flight time over 60 m per condition with relative change per gender.**

| Gender | Normal | 3 kg assisted | 4 kg assisted | 5 kg assisted |
|---|---|---|---|---|
| *Step velocity* † | | | | |
| Women (m/s) | 7.68±0.40 | 7.98±0.29* (+3.9%) | 8.17±0.29* (+6.5%) | 8.35 ±0.30* (+8.8%) |
| Men (m/s) | 8.81±0.30 | 9.00±0.35* (+2.1%) | 9.10±0.37* (+3.3%) | 9.29±0.36* (5.4%) |
| *Step frequency* | | | | |
| Women (Hz) | 4.30±0.20 | 4.26±0.19 (-0.9%) | 4.29±0.24 (-0.2%) | 4.28 ±0.24 (-0.4%) |
| Men (Hz) | 4.38±0.23 | 4.38±0.24 (0.0%) | 4.36±0.24 (-0.6%) | 4.40±0.23 (+0.5%) |
| *Step length* † | | | | |
| Women (m) | 1.80±0.08 | 1.87±0.09* (+3.6%) | 1.90±0.08* (+5.6%) | 1.94 ±0.10* (+7.9%) |
| Men (m) | 2.02±0.08 | 2.06±0.07* (+2.1%) | 2.08±0.10* (+3.3%) | 2.11±0.07* (+4.8%) |
| *Contact time* † | | | | |
| Women (s) | 0.102±0.010 | 0.100±0.082 (-1.2%) | 0.098±0.065 (-3.6%) | 0.098±0.082* (-3.8%) |
| Men (s) | 0.113±0.009 | 0.111±0.011 (-1.3%) | 0.111±0.010 (-1.3%) | 0.110±0.011* (-2.3%) |
| *Flight time* † | | | | |
| Women (s) | 0.133±0.067 | 0.135±0.080 (+1.4%) | 0.136±0.084 (+1.9%) | 0.134±0.073 (+0.9%) |
| Men (s) | 0.119±0.013 | 0.118±0.012 (-0.5%) | 0.118±0.014 (-0.3%) | 0.118±0.012 (-0.7%) |

† indicates a significant difference for this parameter between men and women in every condition on a p<0.05 level.

* indicates a significant difference with the normal 60m condition on a p<0.05 level.

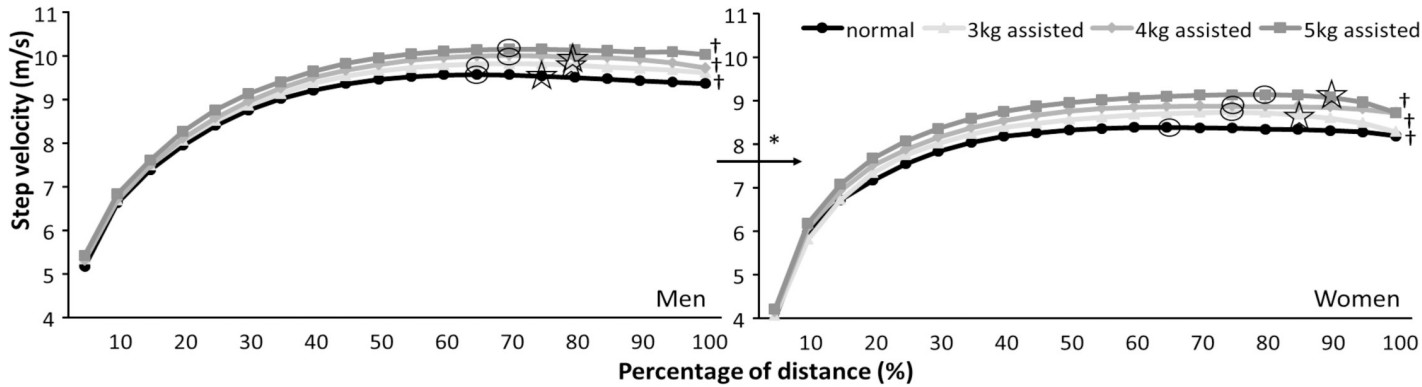

**Fig 3. Average velocity per 5% of the distance during normal and 3, 4, and 5 kg assisted 60 m sprints for men and women.** * indicates a significant difference between men and women for each of the sprint conditions. ⊙ indicates where maximal velocity was reached for this condition. indicates where values decreased again for this condition.

Step length increased significantly for both genders until 45–50% of the distance. From then on it mainly plateaued in all condition in both men and women. Until the step length increased again during the last 5% (Fig 4). Step frequency reached maximal at 0–15% of the distance and decreased again at the last 5% of the distance for both genders.

Contact time decreased until 20–35% of the distance in women and 35–50% of the distance in men where it plateaued for the rest of the distance (Fig 5). Flight time increased until 45–50% of the distance for all conditions in men where it plateaued until the last 5% and increased again. In women flight time increased after 10%, until 35–45% of the distance. It flattened and increased with the 3 kg assisted load in the last 15% of the distance and with the other loads in the last 5% (Fig 5).

## Discussion

The main findings of the present study were that using increasing assisted loads resulted in shorter 60 m times, because of higher step velocity mainly caused by longer step lengths in both genders, and by shorter contact times in women. Men had longer step lengths, longer contact times, and shorter flight times than women. However, the assisted loads had a greater effect on women than on men. These time differences with gender were the result of greater and a longer duration of increase in maximal step velocity with increasing assisted loads for

**Table 4. Maximal (±SD) step velocity and distance at which it is reached and decreases again, in each condition and per gender.**

| Gender | Normal | 3 kg assisted | 4 kg assisted | 5 kg assisted |
|---|---|---|---|---|
| *Step velocity* | | | | |
| Women (m/s) | 8.37±0.39 | 8.73±0.38* | 8.87±0.40* | 9.13 ±0.42* |
| Men (m/s) | 9.57±0.36 | 9.82±0.41* | 10.00±0.43* | 10.14±0.40* |
| *Distance maximal velocity reached* | | | | |
| Women (%) | 65 | 75* | 75* | 80* |
| Men (%) | 65 | 65 | 70* | 70* |
| *Distance decrease maximal velocity* | | | | |
| Women (%) | 100 | 85* | 100 | 90* |
| Men (%) | 75 | 80* | 80* | 100* |

* indicates a significant difference with the normal 60m condition on a p<0.05 level.

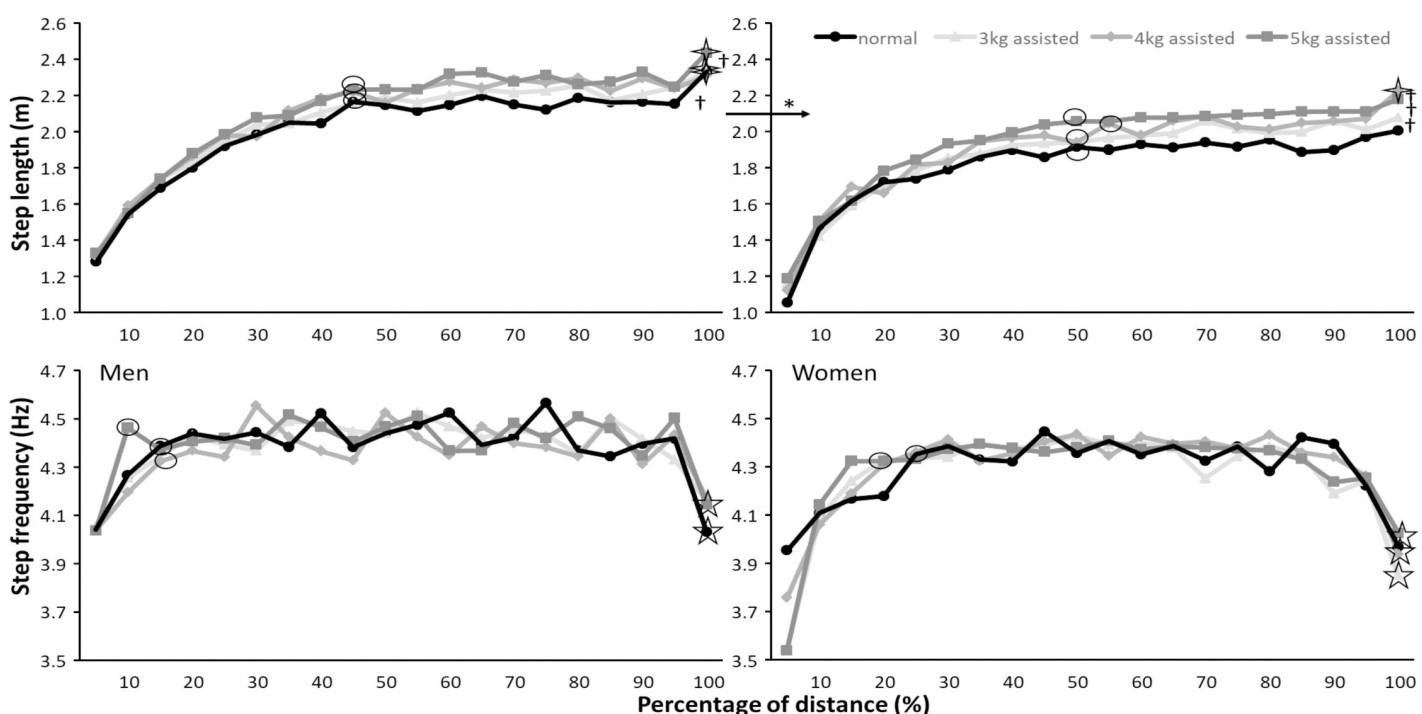

**Fig 4. Average step length and frequency (± SEM) per 5% of the distance during normal and 3, 4, and 5 kg assisted 60 m sprints for men and women.** * indicates a significant difference between men and women for each of the sprint conditions. † indicates a significant difference between these two sprint conditions. ⊙ indicates where maximal/minimal values were reached before plateau for this condition. indicates where maximal values increased for this condition. indicates where values decreased again for this condition.

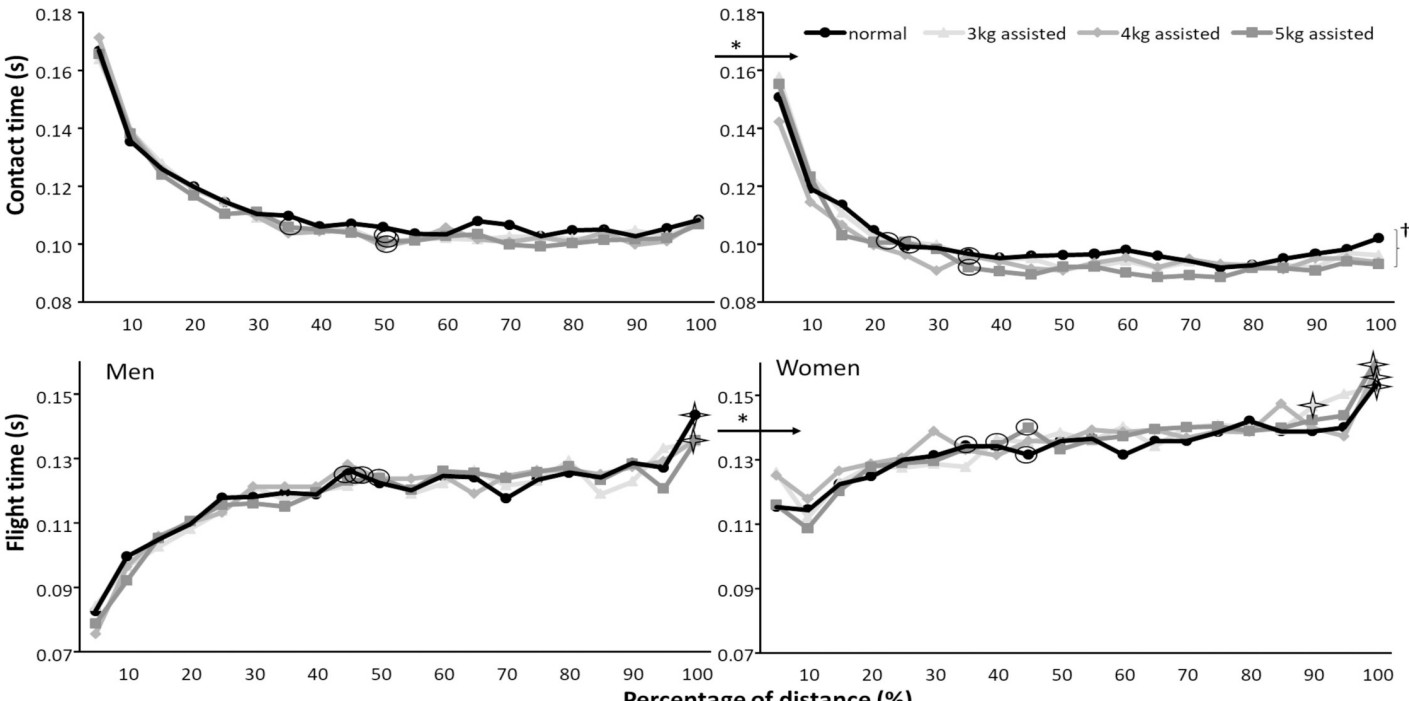

**Fig 5. Average contact and flight times per 5% of the normal and 3, 4, and 5 kg assisted 60 m sprints for men and women.** * indicates a significant difference between men and women for each of the sprint conditions. † indicates a significant difference between these two sprint conditions. ⊙ indicates where maximal/minimal values were reached for this condition. indicates where maximal/minimal values increased again for this condition.

women (70–80% of distance) than men (65–70% of distance). This was mainly caused by shorter contact times, and more increased step lengths in women compared to men.

The kinematics in the present study are in line with the findings of Mero and Komi [9] in elite sprinters. The men have higher step velocity, longer steps or less steps in 60 m (Table 2), longer contact time and shorter flight times, but the same average step frequency compared with the women (Table 3). This could be partially explained by the body height of the men (which was on average 0.14 m (8%) higher) and thereby their longer limb length. As maximal stride length correlates very well with step length (r = 0.59–0.70) [19, 23–25], the taller males had 12% longer step length and thereby on average 4 steps less over 60 m (Table 2) than the females. Beside the anthropometric differences between genders, the longer step length is probably also caused by a lower capacity for women to produce horizontal force at high running velocities [18] as women have a lower leg muscle mass relative to their total body mass than men [26]. In addition, contact and flight times were different between the men and women as men had longer contact time than women, but also shorter flight times, which are also probably caused by the body height and mass differences. The higher body mass and height in males creates a greater moment of inertia, especially in the first steps of the sprint, and when the athletes accelerate this effect will decrease and therefore men decrease contact time over a longer distance than women. Since the flight time in women is higher than in the men, the step frequency is not different between genders (Table 3).

The effect of the assisted load was comparable with earlier studies [10, 15–17, 27, 28] that showed increases in maximal velocity varying from 2.7% [17] to 11.5% [9]. In these studies loads were used from 28 [16] to 50.6 N [17] that resulted in higher maximal velocities. Furthermore, the increases in maximal velocity differ much between the studies, which is caused by the type of pulling system; some used an elastic or rubber rope that reduced in pulling force during the sprints [9, 16], while in others a pulley system that gave constant pulling forces during the whole assisted sprint [10, 15, 17, 28] was used. Furthermore, the level of the athletes and their sprint experience influenced the enhancement of sprint times during the assisted loads. This meant that lower level female and male sprinters had greater advantage (increase in maximal velocity) from the assisted load than elite sprinters [9].

In the present study, the assisted load had a larger effect upon the women (-6.7%) than the men (-4.2%) in 60 m sprint times (Table 1). The changes in time were followed by an increase in maximal step velocity, and by maintaining this step velocity longer as the assisted load is higher. Since the absolute assisted load was the same and the women were lighter than the men the relative propulsive force was higher and therefore could pull women longer to a relatively higher step velocity (increase from 65% in normal run to 85% of total distance with 5 kg, Table 4) than men (from 65% to 70% of total distance, Fig 3, Table 4). This higher step velocity was the result of increased step length with each assisted load for women (+3–7.9%) and for men only between normal and 3 kg (+2.1%) and for 5 kg assisted loads (+4.8%) (Fig 4). However, the distance at which the maximal step length was reached did not change much (45–55% of total distance) with increasing assisted loads. This coincided with reaching the maximal flight times in both men and women. Also, the shortest contact time was reached in men at around 50% of the total distance after which it reached a plateau. This plateau might be indicative of the 'second transition phase' during acceleration as referred to previously by Plamondon and Roy [29] and Nagahara et al. [30]. This transition is characterized by a change from a leaning position to a more stable upright position and thereby termination, in which athletes adopt different lower limb mechanics around this point in a sprint effort in preparation for upright sprinting [30]. This transition is reported to occur at around step 14–16 (28–32 m) during a regular sprint [29, 30]. Based upon the reported step kinematics with the different load it seems that the assisted load does not have much influence on this transition point.

In women the occurrence of the shortest contact time changed from 25 to 35% of the distance and decreased significantly when pulled with 5 kg assisted force. This indicates that the assisted load helped the acceleration phase more in women than in men due to the greater relative assisted load. Since the women also had a longer flight phase the pulling force helped them to increase step length more than the men.

The assisted load had most influence after 15% of the distance (after 9 m) as no significant differences in kinematics were observed in earlier phases. This was probably caused by the positioning of the athletes as they leaned forwards for producing maximal horizontal forces [1, 31, 32], without breaking the steps. That it occurred from this point on (after 15% of distance) is in accordance with the "first transition" during the acceleration phase when sprinting [28–30]. Previous studies have attributed this to the foot beginning to contact the ground in front of center of gravity and the knee joint starting to flex during ground contact [28, 30]. The pulling force attached to the waist could prohibit the athletes from leaning forward more in the first steps and therefore changes in kinematics are not possible in the first steps.

With increasing assisted loads, the maximal step velocity increased relatively more and for longer for women (70–80% of distance) than men (65–70% of distance). After reaching the maximal step velocity, it decreased again at 75% of the distance in normal 60 m sprint conditions in men to no reduction in maximal step velocity with 5 kg assisted loads. In women the maximal step velocity did not decrease in the normal and 4 kg assisted sprints but deceased at 85% and 90% with 3 and 5 kg assisted loads respectively (Fig 3), indicating the absolute assisted load had a different effect on maintaining the maximal step velocity. In men it seems that with 5 kg assisted loads they can maintain their maximal step velocity, while in women when the load is too great, they will decrease their maximal step velocity at the end of the total distance. These differences in development can be explained by more braking during foot placement that occurs during assisted sprints with larger loads [9].

There were some limitations in this current study. Firstly, the results are only investigated as the mean step kinematics for men and women and not the individual adaptations to the changing assisted loads. The individual athletic level of the men and women was not the same as they ranged from 10.27–11.97 s (men) and 11.74–13.07 s (women). Therefore, the strength to adjust with increasing assisted load could be different, besides the anthropometric differences between and within gender. It is possible that the faster sprinters adapt differently to the pulling force than the slower sprinters as observed in the study of Mero and Komi [9] in which the faster male sprinters increased step frequency when performing assisted sprints, while the female sprinters and slower male sprinters increased step length. A possible explanation for these findings is that the athletes were simply not able to increase limb velocity to the degree required to achieve greater step frequency under the assisted sprint condition. Most athletes had some experience with assisted sprint (elastic band, pulley system), but not with this specific system. It is possible that with greater exposure to these different assisted loads, athletes may learn to make the necessary adjustments to allow them to increase step frequency alongside the increases in step length as Mero and Komi [33] observed in elite female sprinters, who increased step frequency with assisted sprints compared to normal sprints. Kristensen et al. [13] observed an increase of step frequency after 6 weeks of assisted sprint training in sport students.

The present investigation did not conduct any direct or indirect assessment of kinetics, such as ground reaction forces or torques generated at each joint segment, or 2- or 3-dimensional joint kinematical analyses. Clearly, the absence of these measurements is a limitation, which could have given more information about acute changes in muscle recruitment, muscle stiffness and possible altered joint kinetics and kinematics [27]. Future investigations should include these measurements of detailed muscle activation, kinetic and/or kinematic analysis of

lower limb joints and limb segments to get a better understanding of the alterations that occur with increasing assisted load. Furthermore, detailed anthropometrics (fat percentage and muscle mass) and muscle strength should be included, which could help in explaining step kinematics changes within and between men and women. In addition, the pulling force should be individualized to investigate if similar enhancements in sprinting times were caused by similar adaptations (step length, step frequency) in men and women and at different experience levels, or if these adaptations are individually different [34].

In conclusion, the assisted loads had a greater effect on women than on men, as shown by their larger decrease in sprint times. These time differences in gender were the result of greater and longer duration increase in maximal step velocity with increasing assisted loads for women (70–80% of distance) than men (65–70% of distance). This was mainly caused by shorter contact times, and more increase in step lengths in women compared to in men. The practical implication for trainers and athletes is that assisted sprints make it possible to reach higher step velocities and hold this maximal step velocity for a longer time when the assisted load is high enough ($> 30$ N). This could be a training impulse to move the speed barrier upwards. However, when the assisted load is too high, step kinematic alterations will not help in maintaining this maximal step velocity at this level, as seen with 5 kg loads in women. Therefore, individual assisted loads should be applied to increase maximal step velocity and keep this velocity, without introducing too much braking in the steps. Based upon the findings of the present investigation, assisted loads of 4–5 kg can be applied depending upon the anthropometrics and experience level of the athlete.

## Supporting information

**S1 Data.**
(PDF)

## Author Contributions

**Conceptualization:** Roland van den Tillaar.

**Data curation:** Roland van den Tillaar.

**Formal analysis:** Roland van den Tillaar.

**Investigation:** Roland van den Tillaar.

**Methodology:** Roland van den Tillaar.

**Validation:** Roland van den Tillaar.

**Visualization:** Roland van den Tillaar.

**Writing – original draft:** Roland van den Tillaar.

**Writing – review & editing:** Roland van den Tillaar.

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
