## [Decision Letter · Decision Letter 0]

29 Mar 2021

PONE-D-21-02963

Comparison of development of Step-Kinematics of Assisted 60 m Sprints with different pulling forces Between Experienced Male and Female Sprinters

PLOS ONE

Dear Dr. van den Tillaar,

Thank you for submitting your manuscript to PLOS ONE. After careful consideration, we feel that it has merit but does not fully meet PLOS ONE’s publication criteria as it currently stands. Therefore, we invite you to submit a revised version of the manuscript that addresses the points raised during the review process.

We look forward to receiving your revised manuscript.

Kind regards,

Dragan Mirkov, Ph.D.

Academic Editor

PLOS ONE

Journal Requirements:

https://onlinelibrary.wiley.com/doi/abs/10.1002/tsm2.130

https://commons.nmu.edu/isbs/vol36/iss1/64/

https://www.tandfonline.com/doi/abs/10.1080/14763141.2018.1442871?journalCode=rspb20

In your revision ensure you cite all your sources (including your own works), and quote or rephrase any duplicated text outside the methods section. Further consideration is dependent on these concerns being addressed.

Additional Editor Comments (if provided):

Reviewers' comments:

Reviewer's Responses to Questions

**Comments to the Author**

1. Is the manuscript technically sound, and do the data support the conclusions?

Reviewer #1: Yes

Reviewer #2: Yes

2. Has the statistical analysis been performed appropriately and rigorously? 

Reviewer #1: Yes

Reviewer #2: Yes

3. Have the authors made all data underlying the findings in their manuscript fully available?

Reviewer #1: No

Reviewer #2: No

4. Is the manuscript presented in an intelligible fashion and written in standard English?

Reviewer #1: Yes

Reviewer #2: Yes

5. Review Comments to the Author

Reviewer #1: GENERAL COMMENTS

The author aimed to explore how different magnitudes of pulling forces influence kinematic variables during 60 m sprint performance in experienced men and women sprinters. Although this is an important question for many practitioners, the study quality needs to be improved significantly in order to be considered for publication. Firstly, the English language is poor, especially in the introduction section (see specific comments below). Secondly, result section is too long, with too many tables and figures (8 in total) and the information overlaps between text, tables and figures. Therefore, I propose merging the information into either tables or figures, or simply creating a single table with most important findings for all kinematic variables and keeping all the figures. Please report only main results in the text. And finally, discussion section carries the most problems. It is simply too long (more than 6 pages, 11 paragraphs) and not well organised. I propose following organisation (or other that will enable more concise discussion): Main findings (paragraph 1), The influence of assisted load on step velocity (paragraph 2), contact and flight time (paragraph 3), step length and frequency (paragraph 4), limitations (paragraph 5), and conclusions (paragraph 6). Please lower down the number of words in discussion.

SPECIFIC COMMENTS

INTRODUCTION

The general organisation of the paragraph and the information provided in each of the paragraphs are very good. However, there is excess of using the pointing words (“this”, “that”, “these”) which decreases the readability of the manuscript. It is essential to rewrite the bellow indicated sentences.

Line 55: remove “this” before assisted since you have not specified any way of providing assisted force until this part.

Lines 56-60: Please rephrase these sentences due to the poor English: “Some pulley systems, such as elastic bands or being pulled by a partner [9] are not easy to control the amount of assistance exactly over the whole sprint distance. Thereby, these systems are often not suited for research purposes in which assisted load must be controlled to know what exact loads are necessary and for the possibility of replicating the study”

Line 68: The reference 16 was not cited correctly in the text (here and throughout of the text).

Line 68: Please put “between 35 and 45 m” in brackets, since it is very confusing for the readers if it stays like this without brackets or signs of interpunction.

Line 69: Please change “to maximal velocity” by “until reaching maximal velocity”

Line 77: Who are “They” at the beginning of the sentence? Please specify.

Line 77-78: Please rewrite the sentence due to the poor English.

Line 78: “This was mainly due to their shorter step length…” What exactly? Be more specific.

Lines 80-82: Which effect? What kind of assisted effect? You need to specify which effect in this sentence.

Lines 89-93: Please rephrase these sentences due to the poor English: “As not much is known about the acute effect of assisted sprints with different loads on these kinematics for men and women, more detailed information about the acute effect of several different loads upon sprinting kinematics for men and women could help in prescribing training programs involving resisted sprints that are specific for gender.”

METHOD

The Design section needs to be improved. For example: Male and female sprinters cannot be “used in…”. They might perform, run, etc. The length of the distance (60m) is repeated several times. You have not stated the type of study design (i.e., Cross-sectional study design in your case).

Why do you report their personal best at 100m in the Participant section? I think it would be of greater value to report their personal best at 60m.

In the second sentence of the Procedure section, you reported three times the word “best”. Please rephrase this sentence.

The picture of DynaSpeed and actual testing setup would be of value here (Line 130).

Did the subjects perform familiarisation session?

Use either subject or participant systematically, but I think that according to the instructions for authors provided by the PlosOne the use of the word Subject is mandatory. Please check this.

Why you used standing start for such a short sprinting discipline?

Lines 141-146: The sentence is too long and difficult to understand. Please rephrase it.

In the study design you are mentioning only assisted sprints, however, in the Statistical analysis you are mentioning that you performed ANOVA for both resisted and assisted sprints. How do you explain this?

RESULTS

Result section is very difficult to be understand. There are too many tables and figures, and the information is repeated in the text as well. Only the most important information needs to stay in the text, while I propose merging the information from the tables and figures, or at least lowering down the information reported in the tables. The quality of the figures needs to be increased, as well as the figure size.

Line 198: Please check part of this sentence: “in both and on step frequency and contact time”

DISCUSSION

The discussion is extremely long (more than 6 pages). Too many different issues are discussed. I propose next organisation: Main findings (paragraph 1), The influence of assisted load on step velocity (paragraph 2), contact and flight time (paragraph 3), step length and frequency (paragraph 4), limitations (5), and conclusions (paragraph 6). Please lower down the number of words used in discussion.

298: This sentence carries no useful information: “Lower pulling forces did not lead to increased sprint velocity.” In which group? In this study? Is it opposite to your findings?

309: Please change “holding” by “maintaining” or other more appropriate word.

Reviewer #2: GENERAL COMMENTS:

Author made a good effort in addressing the effects of pulling forces (assisted training) on step-kinematics of male and female sprinters. The paper is generally well written except at some points that I addressed in specific comments.

SPECIFIC COMMENTS

Abstract

Page 2, Line 40-41 – you state “…which could be a training impulse to move the speed barrier upwards”. This seems to be a bit of an overreach with your results as you did not investigate the effects. Maybe using MAY instead of COULD would solve this.

Introduction

Page 4, Line 73 – Suggestion: Omit “all the” before the “studies” - Studies of van den Tillar…

Page 4, Line 77 – To what study do you refer by the sentence “They found that women achieve peak velocity earlier than men and that this was lower than in men”?

Page 4, Line 73-80 – Improve the clarity of this part of the test as it is not clear who did what, and which reference refers which statement.

Page 4, Line 80-83 – I’m not sure what is the purpose of the sentence: “Mero and Komi (9)…”. I does not seem to fit with the previous one or the following one. Try to make this and the following sentence more generic and more connected to the previous text.

Page 5, Line 96-97 – Shouldn’t you add WOULD after the “step velocity”?

Page 5, Line 98 – The sentence: “This higher maximal velocity is the result of longer step length, shorter contact times, and higher step rate as found in earlier studies [9, 10, 16, 17]” does not seem to belong here. This was supposed to be cleared out earlier in Introduction. Here just clearly state your hypotheses.

Page 5, Line 99-102 – You state: “In addition, it was expected that 60 m times in women would decrease relatively more than in men…” Was this a hypothesis as well?. After that, you continue with: “due to the fact that the pulling force was equal for men and women and therefore women experience more pulling force relative to their body mass”. Shouldn’t this be part of the methods/results or the discussion? Also, this could be mentioned earlier in introduction.

Methods

Design

You did not explain what kind of study design you used.

Page 5, Line 107 – What does “international” refer to? This means that they are competing internationally or they are coming from different countries?

Page 5, Line 109 – I do not think you need to state here which pulley system you used. You will explain it in procedures.

Participants

Were there participants younger than 18 years as men age of males was 22 ± 6 years? If yes, how did you provide the consent from them? Could you add age ranges?

What was the length of training history?

What part of the season was it?

What about avoiding other types of training for 48 hours? Also, what did they do prior these 48 hours as this could also be a sufficient time for the acute effects of supercompensation on performance, depending on the part of the season were in.

Procedures

General comment: Procedures could be explained in more details. Researchers and trainers may have difficulties replicating, thus reducing the applicability and external validity of this study.

Could you structure this part paragraphs instead of single long paragraph for easier reading, better clarity and understanding?

Although I understand why the individualized warmup was performed, a general information about the structure of warmup would be beneficial for the readers of this manuscript. Like this, it seems that 24 different warmups could be performed without any guidance or without any specificity relative to tasks that participants were warming up for. Given that you emphasize that they used individualized warmup does not make it less important, nor does it explain much.

Did you perform the testing as a training session or it was planned within the athletes’ schedule?

Did participants performed familiarization trials?

What was the rest period between the conditions?

Why the 3, 4, and 5 kg trials were not randomized?

Could you provide the reliability and validity of each device used?

Page 6, Line 129-130 – Add more details on how this works or cite the research that explains the procedure in details.

Page 6, Line 135 – I am not sure if I understand how you measured speed. More details may be needed so the study is replicable to a trainer or researcher less familiar with this device.

Results

Page 8, Line 178-181 – This is too long sentence. It would be clearer if you divide it into 2 sentences.

Page 9, Table 1 – This table has not been mentioned in results section, yet it is positioned before the Table 2 that you mentioned.. Indicate significant results in this table, same as you did in other tables. In table title, you used (±SD), but this was not how you used it in the table itself.

Page 10, Line 197 – check for the accuracy of “assistance condition”. Should this be “assisted conditions”?

Page 10, Line 196-200 – This sentence contains too much information. Make it two sentences.

Page 11, Line 218-222 – This should be 2 sentences. It would make it easier to fallow.

Page 11-12, Line 224-225 – “post hoc comparison did not find a significant step frequency difference between conditions in women” – to my understanding, this was already shown in Table 3?

Page 12, Line 226-227 – I suggest replacing “step velocity” with “it”, “more” with “further”, and “increasing” with “larger”.

Page 12, Line 228-230 – This statement should be clearer and more accurate. I’m not so sure what you wanted to say here.

Discussion

Page14, Line 280 – Maybe using “compared” instead “then” women would be more appropriate here.

Page 14, Line 280 – Consider replacing “This is explainable” with “This could be partially explained”. Also, this statement should be referenced.

Page 14, Line 282 – You start the sentence with “As maximal stride length correlates very well with step length (r = 0.59–0.70) [19, 23-25]. How does this relate to the previous sentence or even with the continuance of this very sentence? I am not really sure what you wanted to say. You also continue this sentence with “the taller males had 12% longer step length and thereby on average 4 steps less over 60 m”. Does this mean that taller males would have 12% longer step when compared to shorter males? Also, here you used males, while through the text you used man. These are not the synonyms. In my opinion, using sex (females and males) instead of gender (women and men) is more appropriate for this research as you are investigating differences performance produced by two biological sexes. However, I do not know whether this fits journal requirements and policies.

Page 14, Line 277-284 – Consider making this part better connected with the following of this paragraph. Transition from anthropometrics to muscular potential to exert forces should be smoother and better linked. Although you may not have it in your sample, males and females occasionally may be of the same height. However, it does not necessarily mean that they would perform the same (i.e., have the same step length and velocity).

Page 15, Line 208-309 – Consider replacing “mainly caused” by “followed”. How are you sure that the changes in 60-m sprint time were caused by an increase in maximal step velocity and by holding this step velocity longer and not the other way around. By pulling the athletes, you are actually the one increasing their sprinting speed and then you follow what is going on with the kinematics and dynamics. Whether a repeated application of your conditions produce acute increase the maximal step velocity and prolong it so sprinters become faster was not investigated here.

Page 15, Line 310-313 – Make this argument clearer. From the stated here, I am not sure what are you trying to say.

Page 16, Line 334-335 – Should you add “in earlier phases” at the end of this sentence?

Page 16, Line 335-337 – You state: “This was probably caused by the positioning of the athletes as they leaned forwards for producing maximal horizontal forces [1, 31, 32], without breaking the steps”. How does the positioning explain non-significant effects of pulling forces in the early phase of sprint?

6. PLOS authors have the option to publish the peer review history of their article (what does this mean?). If published, this will include your full peer review and any attached files.

Reviewer #1: No

Reviewer #2: **Yes: **Filip Kukić

---

## [Author Response · Author response to Decision Letter 0]

3 May 2021

Dear editor and reviewers,

We would like to thank the reviewers for doing a great job on improving our manuscript. We have now changed the manuscript according to the comments of the reviewers to improve the readability of the final manuscript. We now think that this manuscript is ready for publication in your journal.

Comments to the reviewers

GENERAL COMMENTS:

Author made a good effort in addressing the effects of pulling forces (assisted training) on step-kinematics of male and female sprinters. The paper is generally well written except at some points that I addressed in specific comments. 

SPECIFIC COMMENTS

Abstract

Page 2, Line 40-41 – you state “…which could be a training impulse to move the speed barrier upwards”. This seems to be a bit of an overreach with your results as you did not investigate the effects. Maybe using MAY instead of COULD would solve this.

Changed it in may

Introduction

Page 4, Line 73 – Suggestion: Omit “all the” before the “studies” - Studies of van den Tillar…

Deleted

Page 4, Line 77 – To what study do you refer by the sentence “They found that women achieve peak velocity earlier than men and that this was lower than in men”?

Slawinski et al. (2017) and Brechue (2011) found this. This is now changed in the text.

Page 4, Line 73-80 – Improve the clarity of this part of the test as it is not clear who did what, and which reference refers which statement.

We have included the names of the references to the text to improve clarity.

Page 4, Line 80-83 – I’m not sure what is the purpose of the sentence: “Mero and Komi (9)…”. I does not seem to fit with the previous one or the following one. Try to make this and the following sentence more generic and more connected to the previous text. 

We fully agree with the reviewer that it does not fit with the previous or following one. We have rewritten it in:

When comparing the effect of assisted sprints upon step kinematics between men and women only Mero and Komi (9) investigated this, … We think it connects the sentences better now.

Page 5, Line 96-97 – Shouldn’t you add WOULD after the “step velocity”? 

Included to the text.

Page 5, Line 98 – The sentence: “This higher maximal velocity is the result of longer step length, shorter contact times, and higher step rate as found in earlier studies [9, 10, 16, 17]” does not seem to belong here. This was supposed to be cleared out earlier in Introduction. Here just clearly state your hypotheses.

We have changed the sentence that it is a clear expected hypothesis based upon the findings from earlier studies.

Page 5, Line 99-102 – You state: “In addition, it was expected that 60 m times in women would decrease relatively more than in men…” Was this a hypothesis as well?. After that, you continue with: “due to the fact that the pulling force was equal for men and women and therefore women experience more pulling force relative to their body mass”. Shouldn’t this be part of the methods/results or the discussion? Also, this could be mentioned earlier in introduction. 

It is also a hypothesis. We have rewritten the next sentence as we tested several assisted loads and explain what we expect between the sexes when using those different loads. We hope these changes are ok for the reviewer.

Methods

Design

You did not explain what kind of study design you used. 

We have deleted the word design to avoid confusion and just wrote what we have done to answer the research question.

Page 5, Line 107 – What does “international” refer to? This means that they are competing internationally or they are coming from different countries?

Some of them were competing at international level and some were also from other countries. However, we have changed it in experienced to avoid confusion.

Page 5, Line 109 – I do not think you need to state here which pulley system you used. You will explain it in procedures. 

We have changed it in a motorized pulley system.

Participants

Were there participants younger than 18 years as men age of males was 22 ± 6 years? If yes, how did you provide the consent from them? Could you add age ranges?

There were four subjects younger than 18 but we have the consent from their parents. We have included this in the methods now. We could add a age range. However, with the standard deviation it is already visible what the main range is and it seems for us a bit weird to write range on age, but not an the rest of the anthropometric variables.

What was the length of training history?

This we did not ask, but all participants had several years of sprint training behind them even the athletes under 18 had years of experience.

What part of the season was it?

It was during the indoor season, a few weeks before the national championships. We have included this to the text now.

What about avoiding other types of training for 48 hours? Also, what did they do prior these 48 hours as this could also be a sufficient time for the acute effects of supercompensation on performance, depending on the part of the season were in. 

The participants were instructed to avoid undertaking any maximal sprint training in the 48 hours prior to testing. We have included this to the text.

Procedures

General comment: Procedures could be explained in more details. Researchers and trainers may have difficulties replicating, thus reducing the applicability and external validity of this study.

Could you structure this part paragraphs instead of single long paragraph for easier reading, better clarity and understanding?

We have split up the paragraph in several and include some more information in the methods part to give the reader more information about the methods.

Although I understand why the individualized warmup was performed, a general information about the structure of warmup would be beneficial for the readers of this manuscript. Like this, it seems that 24 different warmups could be performed without any guidance or without any specificity relative to tasks that participants were warming up for. Given that you emphasize that they used individualized warmup does not make it less important, nor does it explain much. 

As it is an individualized warm-up it is difficult to come with more details, because as you said yourself 24 different warm-ups could be performed. In a sprint competition it is also not expected that everybody does the same warm-up before their sprint. But all the warm-ups can lead to maximal performance for each subject, which is the most important for this study. We have added that the warm-up was around 30-45 minutes long. But as said before it is difficult to specify this, since some used jogging, static stretching, drills, while others did no stretching or dynamic exercises. For the point of the present study it is not so important , we think. 

Did you perform the testing as a training session or it was planned within the athletes’ schedule?

It was planned as a training session that was scheduled within the athletes’ schedule.

Did participants performed familiarization trials? 

The participants did not have familiarization trials, but as already written in the text: All subjects had some experience with assisted sprints, but not with this device. It is difficult to perform a familiarization trial in the same session as this could influence the next run. In another study we had athletes perform a familiarization session, but the sprint results were approximately the same.

What was the rest period between the conditions?

The rest between each trial was 10-12 minutes to avoid fatigue. We have include this to the text now.

Why the 3, 4, and 5 kg trials were not randomized?

We wanted to randomize them. However, since we did not know (no earlier studies on this area), we wanted for the safety of the athletes (avoid possible injuries) start with this order since it was also a few weeks before the nationals. Some trainers were a bit sceptic beforehand. After the session this was gone by the trainers.

Could you provide the reliability and validity of each device used?

I the present study we have not investigated the reliability and validity of the devices used. However, we have investigated the step kinematics between this IMU+laser system with 50 force plates and found that the step kinematics were approximately the same for male sprinters between the systems. We are writing the article now and have not finished all analyses. However, no statistically significant differences between any of the step kinematics between the two systems were found. We can’t refer to this study yet since it is not finished. 

Page 6, Line 129-130 – Add more details on how this works or cite the research that explains the procedure in details.

We have included some more information about the system in the following sentences.

Page 6, Line 135 – I am not sure if I understand how you measured speed. More details may be needed so the study is replicable to a trainer or researcher less familiar with this device. 

The laser measured distance over time which then automatically by the software was calculated in velocity at the different steps. This was all done automatically and since it was a commercial product, they don’t want to share the exact details for the filtering and smoothing of the data. However, the system seems to be accurate since it was compared with a system of 50 force plates (gold standard) in which no significant differences in step kinematics were found between the systems. That study is not finished yet, so we cannot refer to it yet. Sorry. We hope that the reviewer understands that the producer of the system does not give away these details.

Results

Page 8, Line 178-181 – This is too long sentence. It would be clearer if you divide it into 2 sentences.

It is divided in two sentences now.

Page 9, Table 1 – This table has not been mentioned in results section, yet it is positioned before the Table 2 that you mentioned. Indicate significant results in this table, same as you did in other tables. In table title, you used (±SD), but this was not how you used it in the table itself. 

We have now mentioned table 1 in the text. We have already written under the table that all numbers were significantly different between sexes and between conditions. Therefore, we did not included * since this would be at all numbers and thereby creating a chaos in our opinion. 

Page 10, Line 197 – check for the accuracy of “assistance condition”. Should this be “assisted conditions”?

This is changed in assisted conditions

Page 10, Line 196-200 – This sentence contains too much information. Make it two sentences. 

We have split the sentence in two.

Page 11, Line 218-222 – This should be 2 sentences. It would make it easier to fallow. 

Sentence is split in two.

Page 11-12, Line 224-225 – “post hoc comparison did not find a significant step frequency difference between conditions in women” – to my understanding, this was already shown in Table 3?

We found a significant effect of step frequency in women as written in line 208. However, post hoc comparison did show a significant difference between the conditions. This was shown in table 3 and not in figure 4. We have changed this in the text.

Page 12, Line 226-227 – I suggest replacing “step velocity” with “it”, “more” with “further”, and “increasing” with “larger”.

We have changed it according to the comments of the reviewer.

Page 12, Line 228-230 – This statement should be clearer and more accurate. I’m not so sure what you wanted to say here.

We have rewritten the statement to be more accurate.

Discussion

Page14, Line 280 – Maybe using “compared” instead “then” women would be more appropriate here.

We have changed it in “compared with”

Page 14, Line 280 – Consider replacing “This is explainable” with “This could be partially explained”. Also, this statement should be referenced.

We have changed it according to the comment of the reviewer. It is referenced in the following sentence. 

Page 14, Line 282 – You start the sentence with “As maximal stride length correlates very well with step length (r = 0.59–0.70) [19, 23-25]. How does this relate to the previous sentence or even with the continuance of this very sentence? I am not really sure what you wanted to say. You also continue this sentence with “the taller males had 12% longer step length and thereby on average 4 steps less over 60 m”. Does this mean that taller males would have 12% longer step when compared to shorter males? Also, here you used males, while through the text you used man. These are not the synonyms. In my opinion, using sex (females and males) instead of gender (women and men) is more appropriate for this research as you are investigating differences performance produced by two biological sexes. However, I do not know whether this fits journal requirements and policies. 

It refers to the previous sentence and is related to the males compared to the females. We have added this now to the sentence to avoid confusion.

Page 14, Line 277-284 – Consider making this part better connected with the following of this paragraph. Transition from anthropometrics to muscular potential to exert forces should be smoother and better linked. Although you may not have it in your sample, males and females occasionally may be of the same height. However, it does not necessarily mean that they would perform the same (i.e., have the same step length and velocity). 

We agree that with the same anthropometry it is perhaps possible to have the same step kinematics. We have changed the transition sentence a bit to get a better connection to the previous part.

Page 15, Line 308-309 – Consider replacing “mainly caused” by “followed”. How are you sure that the changes in 60-m sprint time were caused by an increase in maximal step velocity and by holding this step velocity longer and not the other way around. By pulling the athletes, you are actually the one increasing their sprinting speed and then you follow what is going on with the kinematics and dynamics. Whether a repeated application of your conditions produce acute increase the maximal step velocity and prolong it so sprinters become faster was not investigated here.

We fully agree with the comment of the reviewer. We increase their velocity, but we don’t know if it will help their sprint capacity in normal runs. That is in the future a study we have planned to conduct.

Page 15, Line 310-313 – Make this argument clearer. From the stated here, I am not sure what are you trying to say. 

The absolute assisted load was the same, while women are lighter and therefore the relative propulsive force was higher for women which resulted in a relatively peak velocity reached later of the total distance. We have rewritten the text and hope it is clear now.

Page 16, Line 334-335 – Should you add “in earlier phases” at the end of this sentence?

This is added to the text now.

Page 16, Line 335-337 – You state: “This was probably caused by the positioning of the athletes as they leaned forwards for producing maximal horizontal forces [1, 31, 32], without breaking the steps”. How does the positioning explain non-significant effects of pulling forces in the early phase of sprint?

As written and explained in the following sentences. The pulling force attached to the waist could prohibit the athletes from leaning forward (positioning) more in the first steps and therefore changes in kinematics are not possible in the first steps.

---

## [Decision Letter · Decision Letter 1]

22 Jun 2021

PONE-D-21-02963R1

Comparison of development of Step-Kinematics of Assisted 60 m Sprints with different pulling forces Between Experienced Male and Female Sprinters

PLOS ONE

Dear Dr. van den Tillaar,

Thank you for submitting your manuscript to PLOS ONE. After careful consideration, we feel that it has merit but does not fully meet PLOS ONE’s publication criteria as it currently stands. Therefore, we invite you to submit a revised version of the manuscript that addresses the points raised during the review process.

We look forward to receiving your revised manuscript.

Kind regards,

Dragan Mirkov, Ph.D.

Academic Editor

PLOS ONE

Journal Requirements:

Reviewers' comments:

Reviewer's Responses to Questions

**Comments to the Author**

1. If the authors have adequately addressed your comments raised in a previous round of review and you feel that this manuscript is now acceptable for publication, you may indicate that here to bypass the “Comments to the Author” section, enter your conflict of interest statement in the “Confidential to Editor” section, and submit your "Accept" recommendation.

Reviewer #1: (No Response)

Reviewer #2: All comments have been addressed

2. Is the manuscript technically sound, and do the data support the conclusions?

Reviewer #1: (No Response)

Reviewer #2: Yes

3. Has the statistical analysis been performed appropriately and rigorously? 

Reviewer #1: (No Response)

Reviewer #2: Yes

4. Have the authors made all data underlying the findings in their manuscript fully available?

Reviewer #1: (No Response)

Reviewer #2: No

5. Is the manuscript presented in an intelligible fashion and written in standard English?

Reviewer #1: (No Response)

Reviewer #2: Yes

6. Review Comments to the Author

Reviewer #1: GENERAL COMMENTS

The author aimed to explore how different magnitudes of pulling forces influence kinematic variables during 60 m sprint performance in experienced men and women sprinters. Although this is an important question for many practitioners, the study quality needs to be improved significantly in order to be considered for publication. Firstly, the English language is poor, especially in the introduction section (see specific comments below). Secondly, result section is too long, with too many tables and figures (8 in total) and the information overlaps between text, tables and figures. Therefore, I propose merging the information into either tables or figures, or simply creating a single table with most important findings for all kinematic variables and keeping all the figures. Please report only main results in the text. And finally, discussion section carries the most problems. It is simply too long (more than 6 pages, 11 paragraphs) and not well organised. I propose following organisation (or other that will enable more concise discussion): Main findings (paragraph 1), The influence of assisted load on step velocity (paragraph 2), contact and flight time (paragraph 3), step length and frequency (paragraph 4), limitations (paragraph 5), and conclusions (paragraph 6). Please lower down the number of words in discussion.

SPECIFIC COMMENTS

INTRODUCTION

The general organisation of the paragraph and the information provided in each of the paragraphs are very good. However, there is excess of using the pointing words (“this”, “that”, “these”) which decreases the readability of the manuscript. It is essential to rewrite the bellow indicated sentences.

Line 55: remove “this” before assisted since you have not specified any way of providing assisted force until this part.

Lines 56-60: Please rephrase these sentences due to the poor English: “Some pulley systems, such as elastic bands or being pulled by a partner [9] are not easy to control the amount of assistance exactly over the whole sprint distance. Thereby, these systems are often not suited for research purposes in which assisted load must be controlled to know what exact loads are necessary and for the possibility of replicating the study”

Line 68: The reference 16 was not cited correctly in the text (here and throughout of the text).

Line 68: Please put “between 35 and 45 m” in brackets, since it is very confusing for the readers if it stays like this without brackets or signs of interpunction.

Line 69: Please change “to maximal velocity” by “until reaching maximal velocity”

Line 77: Who are “They” at the beginning of the sentence? Please specify.

Line 77-78: Please rewrite the sentence due to the poor English.

Line 78: “This was mainly due to their shorter step length…” What exactly? Be more specific.

Lines 80-82: Which effect? What kind of assisted effect? You need to specify which effect in this sentence.

Lines 89-93: Please rephrase these sentences due to the poor English: “As not much is known about the acute effect of assisted sprints with different loads on these kinematics for men and women, more detailed information about the acute effect of several different loads upon sprinting kinematics for men and women could help in prescribing training programs involving resisted sprints that are specific for gender.”

METHOD

The Design section needs to be improved. For example: Male and female sprinters cannot be “used in…”. They might perform, run, etc. The length of the distance (60m) is repeated several times. You have not stated the type of study design (i.e., Cross-sectional study design in your case).

Why do you report their personal best at 100m in the Participant section? I think it would be of greater value to report their personal best at 60m.

In the second sentence of the Procedure section, you reported three times the word “best”. Please rephrase this sentence.

The picture of DynaSpeed and actual testing setup would be of value here (Line 130).

Did the subjects perform familiarisation session?

Use either subject or participant systematically, but I think that according to the instructions for authors provided by the PlosOne the use of the word Subject is mandatory. Please check this.

Why you used standing start for such a short sprinting discipline?

Lines 141-146: The sentence is too long and difficult to understand. Please rephrase it.

In the study design you are mentioning only assisted sprints, however, in the Statistical analysis you are mentioning that you performed ANOVA for both resisted and assisted sprints. How do you explain this?

RESULTS

Result section is very difficult to be understand. There are too many tables and figures, and the information is repeated in the text as well. Only the most important information needs to stay in the text, while I propose merging the information from the tables and figures, or at least lowering down the information reported in the tables. The quality of the figures needs to be increased, as well as the figure size.

Line 198: Please check part of this sentence: “in both and on step frequency and contact time”

DISCUSSION

The discussion is extremely long (more than 6 pages). Too many different issues are discussed. I propose next organisation: Main findings (paragraph 1), The influence of assisted load on step velocity (paragraph 2), contact and flight time (paragraph 3), step length and frequency (paragraph 4), limitations (5), and conclusions (paragraph 6). Please lower down the number of words used in discussion.

298: This sentence carries no useful information: “Lower pulling forces did not lead to increased sprint velocity.” In which group? In this study? Is it opposite to your findings?

309: Please change “holding” by “maintaining” or other more appropriate word.

Reviewer #2: Authors addressed all my comments and suggestions from the first review. Given that I am not a native English speaker, I do not feel competent to evaluate the quality of writing. However, to my level of English, it was understandable and I could easily go through the manuscript. Authors could put some more effort and try to shorten the discussion a bit as it is 11 paragraphs long. Other than that, I am ok with the revision.

7. PLOS authors have the option to publish the peer review history of their article (what does this mean?). If published, this will include your full peer review and any attached files.

Reviewer #1: No

Reviewer #2: No

---

## [Author Response · Author response to Decision Letter 1]

1 Jul 2021

Dear editor and reviewers,

We would like to thank the reviewers for doing a great job on improving our manuscript. We have now changed the manuscript according to the comments of the reviewers to improve the readability of the final manuscript. We now think that this manuscript is ready for publication in your journal.

Comments to the reviewers

Reviewer #1: GENERAL COMMENTS

The author aimed to explore how different magnitudes of pulling forces influence kinematic variables during 60 m sprint performance in experienced men and women sprinters. Although this is an important question for many practitioners, the study quality needs to be improved significantly in order to be considered for publication. Firstly, the English language is poor, especially in the introduction section (see specific comments below). Secondly, result section is too long, with too many tables and figures (8 in total) and the information overlaps between text, tables and figures. Therefore, I propose merging the information into either tables or figures, or simply creating a single table with most important findings for all kinematic variables and keeping all the figures. Please report only main results in the text. And finally, discussion section carries the most problems. It is simply too long (more than 6 pages, 11 paragraphs) and not well organised. I propose following organisation (or other that will enable more concise discussion): Main findings (paragraph 1), The influence of assisted load on step velocity (paragraph 2), contact and flight time (paragraph 3), step length and frequency (paragraph 4), limitations (paragraph 5), and conclusions (paragraph 6). Please lower down the number of words in discussion.

SPECIFIC COMMENTS

INTRODUCTION

The general organisation of the paragraph and the information provided in each of the paragraphs are very good. However, there is excess of using the pointing words (“this”, “that”, “these”) which decreases the readability of the manuscript. It is essential to rewrite the bellow indicated sentences.

We have changed this in the introduction.

Line 55: remove “this” before assisted since you have not specified any way of providing assisted force until this part.

We have deleted this from the text.

Lines 56-60: Please rephrase these sentences due to the poor English: “Some pulley systems, such as elastic bands or being pulled by a partner [9] are not easy to control the amount of assistance exactly over the whole sprint distance. Thereby, these systems are often not suited for research purposes in which assisted load must be controlled to know what exact loads are necessary and for the possibility of replicating the study”

We have rewritten it in: Some pulley systems, such as elastic bands or being pulled by a partner [9] cannot easy control the amount of assistance exactly over the whole sprint distance. Thereby, these systems are not well suited for research purposes in which assisted load must be controlled to know what exact loads are necessary and for the possibility of replicating the study.

Line 68: The reference 16 was not cited correctly in the text (here and throughout of the text).

This is changed now in the entire manuscript.

Line 68: Please put “between 35 and 45 m” in brackets, since it is very confusing for the readers if it stays like this without brackets or signs of interpunction.

We have included brackets around them.

Line 69: Please change “to maximal velocity” by “until reaching maximal velocity”

Changed now.

Line 77: Who are “They” at the beginning of the sentence? Please specify.

We have included the references.

Line 77-78: Please rewrite the sentence due to the poor English.

We have changed the sentence in: Slawinski et al. (18) and Brechue (19) found that women achieve peak velocity during a sprint earlier than men and that the peak velocity was lower than in men

Line 78: “This was mainly due to their shorter step length…” What exactly? Be more specific.

We have changed it in: This earlier occurrence and lower peak velocity was mainly due to their shorter step

Lines 80-82: Which effect? What kind of assisted effect? You need to specify which effect in this sentence.

We have changed this in: When comparing the effect of assisted sprints upon step kinematics between men and women only Mero and Komi (9) investigated this, when …

Lines 89-93: Please rephrase these sentences due to the poor English: “As not much is known about the acute effect of assisted sprints with different loads on these kinematics for men and women, more detailed information about the acute effect of several different loads upon sprinting kinematics for men and women could help in prescribing training programs involving resisted sprints that are specific for gender.”

We have changed the sentence in: As not much is known about the acute effect of assisted sprints with different loads on step kinematics for men and women, more detailed information is needed. The acute effect of several different loads upon sprinting step kinematics for men and women could help in prescribing training programs involving assisted sprints that are specific for gender.

METHOD

The Design section needs to be improved. For example: Male and female sprinters cannot be “used in…”. They might perform, run, etc. The length of the distance (60m) is repeated several times. You have not stated the type of study design (i.e., Cross-sectional study design in your case).

We have changed it in: a cross-sectional study design was used in which 24 experienced male and female sprinters performed a normal …

Why do you report their personal best at 100m in the Participant section? I think it would be of greater value to report their personal best at 60m.

All athletes had given their recent 100m pb and not all had this for the 60m indoor times. Furthermore, 100m pb are much better to compare with other studies in which these are often given and not 60m times. 

In the second sentence of the Procedure section, you reported three times the word “best”. Please rephrase this sentence.

We have delete some “best” and changed it in greatest at other places.

The picture of DynaSpeed and actual testing setup would be of value here (Line 130).

We have included a picture of the Dynaspeed and actual setup.

Did the subjects perform familiarisation session?

They did not perform a familiarization session, since they were familiar with the method, but not with this device. This is already mentioned in the text.

Use either subject or participant systematically, but I think that according to the instructions for authors provided by the PlosOne the use of the word Subject is mandatory. Please check this.

It is checked in the manuscript and changed in subject where necessary.

Why you used standing start for such a short sprinting discipline?

We used a standing start, since it is difficult to use a start block start, when there is a line pulling you around your waist. You cannot lean forwards as much, because then the line is in your face.

Lines 141-146: The sentence is too long and difficult to understand. Please rephrase it.

We have changed it in: Contact and flight times throughout the run were derived from an identified running pattern (Figure 1) using wireless 9 degrees of freedom inertial measurement units (IMU) integrated with a 3-axis gyroscope attached on top of the shoelaces of the spikes of each foot. Sampling rate of the gyroscope was 500 Hz with maximal measuring range of 2000°·s-1 ± 3% (Ergotest Technology AS, Langesund, Norway). This running pattern was determined in an unpublished pilot study that compared contact and flight time data measured with an infrared contact mat over 30 m (Ergotest Technology AS, Langesund, Norway) using the patterns of the angular velocity of plantar flexion/extension (ICC=0.94).

In the study design you are mentioning only assisted sprints, however, in the Statistical analysis you are mentioning that you performed ANOVA for both resisted and assisted sprints. How do you explain this?

Writing mistake. It is changed in assisted.

RESULTS

Result section is very difficult to be understand. There are too many tables and figures, and the information is repeated in the text as well. Only the most important information needs to stay in the text, while I propose merging the information from the tables and figures, or at least lowering down the information reported in the tables. The quality of the figures needs to be increased, as well as the figure size.

In this study much interesting data was gathered and important to show. We have tried to delete parts of the results, but it is difficult to delete something without missing the point of the result. The quality of the figures was according to the journal standards with the appropriate figure size. Since it are tiff files it seems that the quality is poor (own experience as reviewer). However, in the final article the quality is good again. Sorry for that.

Line 198: Please check part of this sentence: “in both and on step frequency and contact time”

We have split the sentence in two to make it more readable. It is rewritten in: … while the assisted conditions influenced velocity and step length (F ≥ 9.8, p < 0.001, η2 ≥ 0.39) in both genders. On step frequency and contact times only a significant effect in women (F ≥ 3.5, p ≤ 0.031, η2 ≥ 0.34) was found, but no significant effect on flight time (F = 0.45, p = 0.72, η2 = 0.04, Table 3).

DISCUSSION

The discussion is extremely long (more than 6 pages). Too many different issues are discussed. I propose next organisation: Main findings (paragraph 1), The influence of assisted load on step velocity (paragraph 2), contact and flight time (paragraph 3), step length and frequency (paragraph 4), limitations (5), and conclusions (paragraph 6). Please lower down the number of words used in discussion.

We have tried to cut down the discussion, which was difficult. We think the construction of paragraphs: 1) main findings 2) differences men women, 3) effect of assisted loads general 4) effect of assisted loads between gender 5) limitations 6) conclusion. Since this is the first study that performed such a detailed step kinematics analysis for both men and women, we think it would not be good to take out too much of the discussion to avoid speculations. We hope that the reviewer appreciates our point of view.

298: This sentence carries no useful information: “Lower pulling forces did not lead to increased sprint velocity.” In which group? In this study? Is it opposite to your findings?

It is deleted now.

309: Please change “holding” by “maintaining” or other more appropriate word.

Changed in maintaining

Reviewer #2: Authors addressed all my comments and suggestions from the first review. Given that I am not a native English speaker, I do not feel competent to evaluate the quality of writing. However, to my level of English, it was understandable and I could easily go through the manuscript. Authors could put some more effort and try to shorten the discussion a bit as it is 11 paragraphs long. Other than that, I am ok with the revision.

We have shortened the discussion a little bit. Thank you for reviewing the manuscript.

---

## [Decision Letter · Decision Letter 2]

14 Jul 2021

Comparison of development of Step-Kinematics of Assisted 60 m Sprints with different pulling forces Between Experienced Male and Female Sprinters

PONE-D-21-02963R2

Dear Dr. van den Tillaar,

We’re pleased to inform you that your manuscript has been judged scientifically suitable for publication and will be formally accepted for publication once it meets all outstanding technical requirements.

Kind regards,

Dragan Mirkov, Ph.D.

Academic Editor

PLOS ONE

Additional Editor Comments (optional):

Reviewers' comments:

Reviewer's Responses to Questions

**Comments to the Author**

1. If the authors have adequately addressed your comments raised in a previous round of review and you feel that this manuscript is now acceptable for publication, you may indicate that here to bypass the “Comments to the Author” section, enter your conflict of interest statement in the “Confidential to Editor” section, and submit your "Accept" recommendation.

Reviewer #1: All comments have been addressed

Reviewer #2: All comments have been addressed

2. Is the manuscript technically sound, and do the data support the conclusions?

Reviewer #1: Yes

Reviewer #2: Yes

3. Has the statistical analysis been performed appropriately and rigorously? 

Reviewer #1: Yes

Reviewer #2: Yes

4. Have the authors made all data underlying the findings in their manuscript fully available?

Reviewer #1: Yes

Reviewer #2: Yes

5. Is the manuscript presented in an intelligible fashion and written in standard English?

Reviewer #1: Yes

Reviewer #2: Yes

6. Review Comments to the Author

Reviewer #1: The authors adressed all my comments adequatly, well done. In my opinion the article is now ready for acceptance.

Reviewer #2: I have no more comments. Authors addressed all my suggestions. I congratulate authors on good work.

7. PLOS authors have the option to publish the peer review history of their article (what does this mean?). If published, this will include your full peer review and any attached files.

Reviewer #1: No

Reviewer #2: No

---

## [Editor Report · Acceptance letter]

19 Jul 2021

PONE-D-21-02963R2 

Comparison of development of Step-Kinematics of Assisted 60 m Sprints with different pulling forces Between Experienced Male and Female Sprinters 

Dear Dr. van den Tillaar:

I'm pleased to inform you that your manuscript has been deemed suitable for publication in PLOS ONE. Congratulations! Your manuscript is now with our production department. 

Kind regards, 

on behalf of

Dr. Dragan Mirkov 

Academic Editor

PLOS ONE